# Surgical Clipping of Intracranial Aneurysms Using a Transcranial Neuroendoscopic Approach

**DOI:** 10.3390/brainsci13111544

**Published:** 2023-11-02

**Authors:** Zhiyang Li, Pan Lei, Qiuwei Hua, Long Zhou, Ping Song, Lun Gao, Silei Zhang, Qiang Cai

**Affiliations:** 1Department of Neurosurgery, Renmin Hospital of Wuhan University, Wuhan 430060, China; 2018283020174@whu.edu.cn (Z.L.); 2021283020190@whu.edu.cn (P.L.); 2022203020026@whu.edu.cn (Q.H.); longzhou520@whu.edu.cn (L.Z.); songping1201@163.com (P.S.); lungao@whu.edu.cn (L.G.); 2Department of Neurosurgery, Xiantao First People’s Hospital Affiliated to Changjiang University, Xiantao 433000, China; zhangsilei0622@163.com

**Keywords:** intracranial aneurysm, transcranial neuroendoscopic approach, clip, microneurosurgery

## Abstract

Objective: This retrospective study was performed to evaluate the feasibility and safety of surgically clipping intracranial aneurysms using a transcranial neuroendoscopic approach. Methods: A total of 229 patients with cerebral aneurysms were included in our study, all of whom were treated with clamping surgery at Wuhan University People’s Hospital. They were divided into neuroendoscopic and microscopic groups, according to whether or not neuroendoscopy was used for the clamping surgery. We statistically analyzed the patients’ baseline data, surgical outcomes, and complications, which were then evaluated to assess the treatment effect. Results: The baseline characteristics were not statistically significant, except for gender, for which the proportions of female patients in the two groups were 69 (56.1%) and 46 (43.4%). There were no patients with incomplete aneurysm clamping or parent vessel occlusion in the neuroendoscopic group, and there were 4 (3.8%) and 2 (1.9%) in the microscopic group, respectively; however, there was no statistically significant difference in the comparison of the two groups. The mean operative times of the two groups were 181 min and 154 min, respectively, and were statistically different. However, the mRS scores of the two groups showed no significant difference in patient prognosis. The differences in complications (including limb hemiplegia, hydrocephalus, vision loss, and intracranial infection) were not statistically significant, except for cerebral ischemia, for which the proportions of patients in the two groups were 8 (6.5%) and 16 (15.1%). Conclusions: Neuroendoscopy can provide clear visualization and multi-angle views during aneurysm clipping, which is helpful for ensuring adequate clipping and preventing complications.

## 1. Introduction

Intracranial aneurysm (IA) is one of the most common causes of subarachnoid hemorrhage (SAH), and affects 0.02% to 9% of people [1]. There are two approaches for the treatment of intracranial aneurysm (IA), interventional therapy and surgical clipping, both of which have their advantages and disadvantages in terms of treatment efficacy. Many studies have shown that interventional therapy has fewer postoperative complications than craniotomies [2,3,4]. Unable to afford the high cost of interventional therapy, some patients choose to undergo craniotomy in China.

Intracranial aneurysms carry a risk of hemorrhage related to their size, with larger aneurysms at higher risk for rupture [5]. The aim of microsurgical clipping is to achieve complete occlusion of the aneurysm while preserving blood flow in the involved parent artery, its branches, and perforators, with minimal brain tissue trauma [6]. The introduction of the microscope to aneurysm surgery in the 1960s tremendously improved outcomes; however, even experienced surgeons must deal with inadequately clipped aneurysms. Incomplete aneurysmal occlusion, unexpected branch occlusion, and partial vessel occlusion with significant blood flow reduction are frequently reported due to difficulty visualizing the aneurysmal sac because it can be obscured from microscopic view [6,7,8]. Neuroendoscopy has been applied to the treatment of a variety of neurosurgical diseases, including ventricular cerebral pool disorders, skull base disorders, intracranial aneurysms, spinal cord disorders, microvascular decompression, cerebrospinal fluid leakage, and intracranial hematomas [9,10]. Certain features of neuroendoscopic techniques help to overcome the shortcomings of microsurgical aneurysm clipping.

In 2006, Kassam et al. first reported a purely endonasal endoscopic approach (EEA) for clipping coiled vertebral artery aneurysms [11]. Several other case reports have followed, demonstrating the feasibility of this technique for various skull base aneurysms. Gardner reported 11 intracranial aneurysms arising from the paraclinoid ICA, posterior cerebral artery, and basilar apex in 10 patients who underwent EEA clipping [12]. All patients recovered well, and none required additional surgical or endovascular intervention or experienced postoperative neurologic deterioration. However, a systematic review about the safety and effectiveness of endoscopic endonasal intracranial aneurysm clipping identified complications related to the procedure in approximately 26% of cases, severe ischemic complications occurred in 15% of cases, and 83% of aneurysms were clipped successfully [13]. Moreover, their review highlights the lack of good quality studies that truly assess the safety and effectiveness of the technique. The aims of our study were to evaluate the safety and efficacy of neuroendoscopy in the microclipping of intracranial aneurysms.

## 2. Patients and Methods

### 2.1. Patient Population

This retrospective study included a total of 229 patients who underwent intracranial aneurysm clamping between January 2017 and January 2023 at the Department of Neurosurgery, Wuhan University People’s Hospital, China. A total of 123 patients who underwent endoscope-assisted microneurosurgery (EAM, *n* = 28) or full neuroendoscopy (FNE, *n* = 95) were defined as the endoscopic group. The other 106 patients who underwent microscopic surgery without neuroendoscopy were defined as the microscopic group. The inclusion criteria were as follows: (1) recurring headache; (2) CT scan showing presence of SAH or not; (3)DSA-confirmed diagnosis of intracranial aneurysm; (4) aneurysm diameter of 5~20 mm. The exclusion criteria were as follows: (1) complete calcification of the aneurysm wall making surgical clamping impossible; (2) SAH with Hunt and Hess grade of V or GCS score ≤ 5; (3) secondary intracranial aneurysms, such as traumatic aneurysms and infectious aneurysms; (4) intracranial aneurysms complicated with other cerebrovascular malformations; (5) patient refusal of surgery; (6) incomplete records (Figure 1).

The study protocol was approved by the Ethics Committee of the People’s Hospital of Wuhan University (WDRM2022-KS002). The requirement for obtaining informed consent from the patients was waived because the study is based on information that is part of routine clinical care and medical records. All performed procedures followed the 1964 Helsinki declaration and its later amendments or comparable ethical standards.

### 2.2. Data Collection

The patients’ clinical information, evaluated and collected from the hospital information system, was as follows: (1) demographic characteristics, including gender and age; (2) aneurysm morphology, including aneurysm location and aneurysm diameter; (3) admission clinical grading, including Hunt–Hess grade; (4) surgical conditions, including surgical procedure time, inadequately clipped aneurysms, parent vessel occlusion, and number of bleeds prior to surgery; (5) patient prognosis, including mRS score at three months and limb hemiplegia; (6) complications, including hydrocephalus, intracranial infection, vision loss, and cerebral ischemia.

A diagnosis of hydrocephalus was identified using the following standards: (1) Evans index > 0.3 (the ratio of the greatest distance between the bilateral anterior horns of the lateral ventricles and the greatest internal distance of the skull) and (2) an enlarged anterior horn of lateral ventricle, temporal horn, and third ventricle with periventricular edema [14].

Cerebral ischemia requires one of the following criteria to be met: (1) no other underlying causes of neurologic deficits in the postoperative period; (2) Glasgow Coma Scale single score or a decrease of ≥2 points in the total score; (3) new hypodense lesions observed on CT scan [15].

### 2.3. Surgical Procedures

During the endoscopic group’s procedure, the neuroendoscope was held by an assistant and the endoscope was placed slowly and smoothly around the aneurysm at an appropriate angle and diameter. We observed the local anatomy of the aneurysm in relation to the aneurysm-carrying artery and surrounding structures. A side-angled endoscope was used to visualize the structures below the aneurysm and the surgical dead space (e.g., ophthalmic segment of the internal carotid artery, etc.). An appropriate aneurysm clip was selected to clip the aneurysm and observed for complete closure. At the same time, a microscope was prepared for all patients in case of intraoperative aneurysm rupture with excessive bleeding, and in that case, we immediately switched to microscopic surgery. Surgical procedures within the microscopic group were exclusively performed utilizing a microscope without the neuroendoscope.

Aneurysms were exposed through the traditional approach or a modified approach. For the traditional approach, usually, the pterional or supraorbital keyhole approach was used for ACoA (Figure 2), the pterional approach was used for the beginning of A2 segment aneurysms (Figure 3), the pterional or extended pterional approach was used for PCoA or MCA aneurysms (Figure 4, Figure 5 and Figure 6), and the far-lateral surgical approach was used for PICA (Figure 7). During the craniotomy, the anterior clinoid process (ACP) was removed for most PCoAs and part MCAs, which we had reported previously. For the modified approach, we usually chose a small incision line close to the location of the aneurysm or hematoma (Figure 8).

### 2.4. Statistical Analysis

Kolmogorov–Smirnov tests were used to test all clinical covariates for normal distribution before statistical analysis. Continuous variables are summarized as interquartile ranges or means ± standard deviation, and categorical variables are presented as frequencies (percentages). Continuous variables were compared via a Mann–Whitney U test or independent *t* test, whereas categorical variables were tested via a χ2 test or Fisher’s exact test. A *p* value < 0.05 was considered to be statistically significant.

## 3. Results

### 3.1. Baseline Characteristics

In the neuroendoscopic group and microscopic group, the mean ages were 58.5 and 59.5 years, and there were 69 (56.1%) and 46 (43.4%) female patients, respectively, with statistically significant differences in gender. MCA, PCoA, ACoA, A2-A4, PICA, and Opht were 40 (32.5%), 38 (30.9%), 27 (22%), 10 (8.1%), 5 (4.1%), and 3 (2.4%) in the neuroendoscopy-assisted group; and 29 (27.4%), 34 (32.1%), 22 (20.8%), 9 (8.5%), 6 (5.7%), and 6 (5.7%) in the microscopic group. The mean aneurysm diameters of the two groups were 9.15 mm and 5.51 mm; however, there was no difference in the location of the aneurysm or the display of the aneurysm between the two groups. In addition, there was no statistical difference in the Hunt–Hess grade and number of bleeding instances between the two groups. Table 1 shows the baseline characteristics of the patients.

### 3.2. The Surgical Efficacy and Complications

There were no patients with incomplete aneurysm clamping or parent vessel occlusion in the neuroendoscopic group, and there were 4 (3.8%) and 2 (1.9%) in the microscopic group, respectively; however, there was no statistically significant difference in the comparison of the two groups. The mean operative times of the two groups were 181 min and 154 min, respectively, and were statistically different. However, the mRS scores of the two groups showed no significant difference in patient prognosis. Differences in complications (including limb hemiplegia, hydrocephalus, vision loss, and intracranial infection) were not statistically significant except for cerebral ischemia, for which the proportions of patients in the two groups were 8 (6.5%) and 16 (15.1%), respectively. Table 2 shows the complications and prognoses of the patients.

## 4. Discussion

Microsurgical clipping is one of the most effective treatments for intracranial aneurysms; the goal of this surgery is to achieve complete occlusion of the aneurysm while preserving blood flow in the involved parent artery, its branches, and perforators, with minimal brain tissue trauma. However, even experienced surgeons must deal with inadequately clipped aneurysms. Incomplete aneurysmal occlusion, unexpected branch occlusion, and partial vessel occlusion with significant blood flow reduction are frequently reported due to difficulty visualizing the aneurysmal sac because it can be obscured from microscopic view. Our study showed that parent vessel occlusion and inadequately clipped aneurysm occurred in 2 (1.9%) and 4 (3.8%) patients in the microscopy group, respectively, and then, these complications did not occur in the endoscopic assisted group. To overcome the intrinsic limits of microscopy, the endoscope was first used during the microsurgical resection of pituitary lesions by Apuzzo and Halve in the late 1970s [16,17]. It optimized visualization of the neurovascular structures located beyond the line of a microscope’s sight. This technique ultimately expanded into endoscope-assisted microneurosurgery (EAM), which was introduced by Matula in 1995 to treat posterior fossa and parasellar region lesions [18].

Other techniques mainly included the transcranial and endonasal approaches that were developed following the technological maturation of endoscopic surgery. Axel Perneczky employed limited keyhole approaches to treat different intracranial pathologies, including aneurysms [19]. Initially, the endoscopes were used before and after clipping. It is more frequently used for real transcranial endoscope-controlled clipping procedures because it increases light intensity in the surgical field and augments the visual field [20]. In the 1990s, endonasal approaches were discarded for aneurysms mainly due to a narrow surgical corridor, limitations for skull base reconstruction, and frequent CSF leakage [21]. Fortunately, these shortcomings were addressed by the development of the mucosal nasal septal flap and novel skull base surgery techniques in the 2000s. In 2006, Kassam et al. first reported a purely endonasal endoscopic approach (EEA) for clipping coiled vertebral artery aneurysms [11]. Several other case reports have followed, demonstrating the feasibility of this technique for various skull base aneurysms. In 2011, Froelich reported the first true EEA for clipping an ACoA aneurysm [22].

Despite improvements in techniques and instrumentation, the main downsides of clipping aneurysms through EEA are the narrow surgical corridor, difficulty controlling the parent vessel, and the inability to perform vascular bypasses [19]. The strongest argument against the EEA for aneurysm clipping may be based on the fact that transcranial approaches are safe, effective, and well studied, and have low morbidity and mortality rates. The ability to manage major vessels and catastrophic hemorrhagic complications are also major challenges for EEA use in aneurysm surgery [23]. Nevertheless, EEA may be useful in rare, well-selected cases for ventrally and medially situated aneurysms where it provides better visualization, increased or equivalent vascular control, and a more direct surgical corridor than transcranial approaches [23]. However, an intrinsic limit of EEA is that it cannot easily overcome postoperative CSF leakage. This complication occurred in three patients in Gardner’s series [12], and a review by Jonathan found that patients undergoing EEA clipping of posterior circulation aneurysms had a significantly higher rates of postoperative CSF leakage (*p* = 0.047, Fisher’s Exact Test) compared to patients with anterior circulation aneurysms [23].

Although a purely transcranial approach is relatively new for aneurysm clipping, CSF leakage has not been reported. Perneczky clipped seven aneurysms with exclusive use of the endoscope through a pterional craniotomy opening, and Radovanovic performed endoscopic aneurysm clipping through a 2 × 2 cm craniotomy with good results and no complications. In our series, all patients were successfully treated without CSF leakage, morbidity, or aneurysm rupture directly attributable to neuroendoscopy. Certain features of neuroendoscopic techniques help to overcome the shortcomings of microsurgical aneurysm clipping: (1) The light source is close to the surgical field and offers better illumination; (2) it provides a clear, close-up view of the anatomy; and (3) it gives a multi-angled view of the region. These features help the surgeon understand the neuro-vascular anatomy and the relationship between the aneurysm(s) and surrounding structures, which could reduce unwanted outcomes. On the other hand, some disadvantages of neuroendoscopy have been reported: (1) initial neuroendoscopic inspection might cause a rupture, (2) three-dimensional views are not possible, (3) it is useless when there is blood in the operative field, and (4) there is limited instrumentation specifically designed for neuroendoscopic surgery [24].

Wide-neck aneurysms and narrow-neck aneurysms lead to different effects of the aneurysm neck on rupture risk due to differences in hemodynamics. Some studies found that high AR was positively correlated with aneurysm rupture [25]. New research found a non-linear correlation of AR with IA rupture in the AR range of 1.08–1.99; additionally, the prevalence of IA rupture was lower for the increment of AR, the prevalence of IA rupture increased in the AR range of 3.42–4.08, and there was no association between AR and the prevalence of IA rupture in the AR range of 1.99–3.42 [26]. The multi-angled view afforded by the endoscope has improved the visualization of perforating arteries, especially those arising from the posterior wall of the parent artery, so it has gradually been applied in aneurysm surgery. Understanding the relationship between the aneurysm and surrounding structures before, during, and after aneurismal clipping could help ensure adequate aneurysmal clipping [27,28].

## 5. Conclusions

In conclusion, neuroendoscopy can provide clear visualization and multi-angle views during aneurysm clipping and a good view of the aneurysm neck and the perforating arteries in the blind spot of the operative field, and it helps to assess the effectiveness of aneurysm clamping. Neuroendoscopy can help to ensure the success and safety of cerebral aneurysm clamping.

## Figures and Tables

**Figure 1 brainsci-13-01544-f001:**
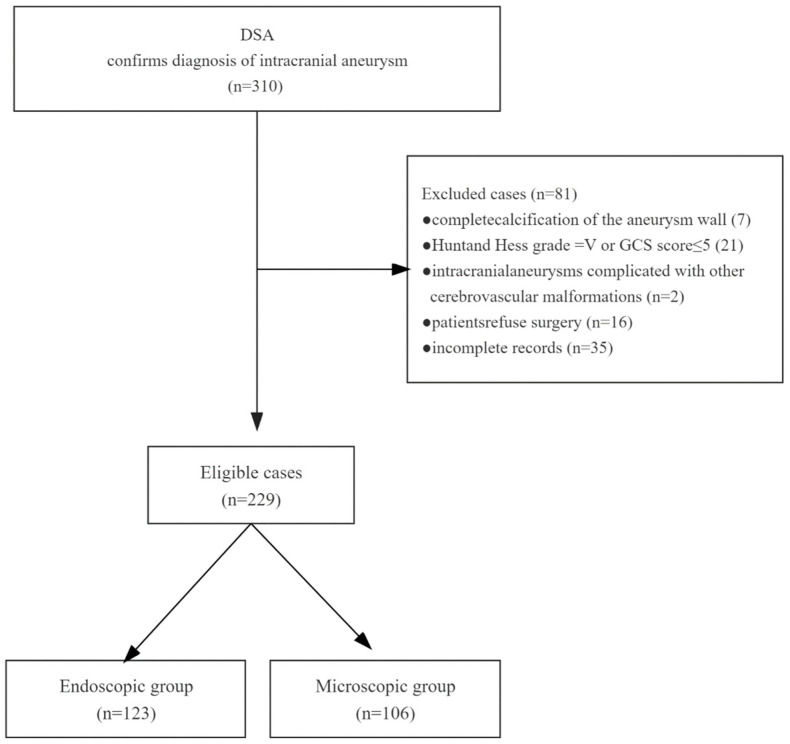
Flow chart for patient selection. Information on 310 patients was collected, of which 81 patients were excluded due to multiple reasons, and finally, 229 patients were included for further analysis.

**Figure 2 brainsci-13-01544-f002:**
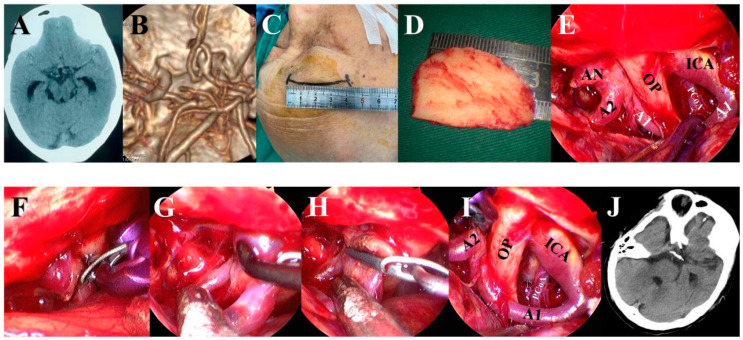
An ACoA aneurysm was clipped via the traditional supraorbital keyhole approach under the neuroendoscope. (**A**) Preoperative CT showed mild subarachnoid hemorrhage. (**B**) Pre-operative CTA scan showed a saccular aneurysm at the anterior communicating artery complex. (**C**) Supraorbital keyhole approach was used. (**D**) The diameter of the bone flap was about 3 cm. (**E**) Sylvian fissure was sharply dissected and ICA, MCA, segment A1, and optic nerve were exposed under the neuroendoscope. (**F**) Aneurysm was exposed and clipped under the neuroendoscope. (**G**) We checked whether the parent artery and its branches were clamped correctly. (**H**) The aneurysm was cut to confirm complete clipping. (**I**) We carefully checked the operation area before closing the dura matter. (**J**) Postoperative CT scan showing that the clip of the aneurysm had proper placement.

**Figure 3 brainsci-13-01544-f003:**
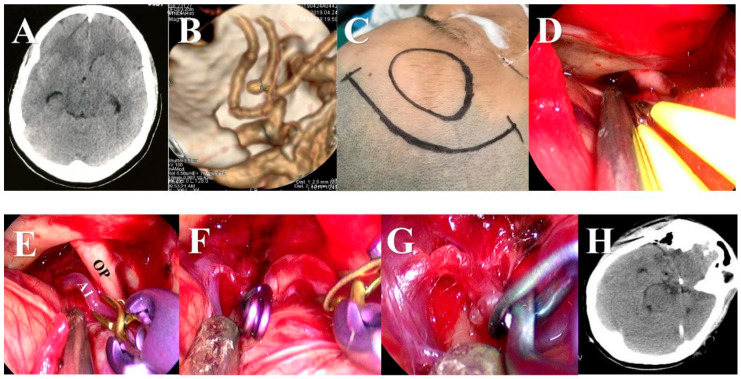
An A2 segment aneurysm was clipped via the traditional pterional approach under the neuroendoscope. (**A**) Preoperative CT showed subarachnoid hemorrhage. (**B**) Preoperative CTA showed that the aneurysm was located at the beginning of the A2 segment. (**C**) Pterional approach was used. (**D**) We opened the arachnoid membrane at the base of the anterior cranial fossa and released cerebrospinal fluid. (**E**) The right A1 segment was exposed and clipped temporarily. (**F**) Aneurysm was exposed and clipped under the neuroendoscope. (**G**) The parent artery and its branches were checked carefully. (**H**) Postoperative CT scan showing that the clip of the aneurysm had proper placement.

**Figure 4 brainsci-13-01544-f004:**
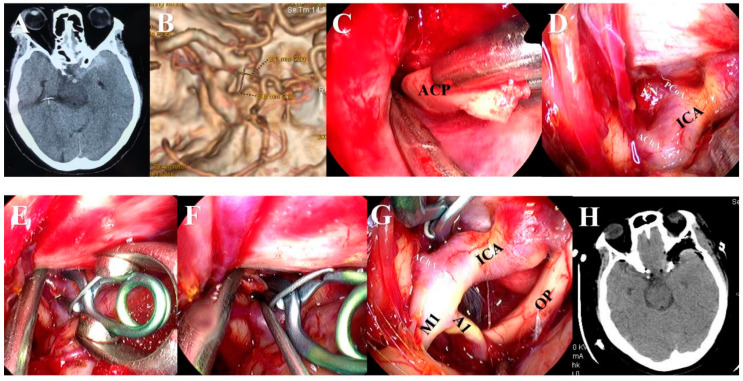
A ruptured PCoA aneurysm was clipped via the traditional pterional approach under the neuroendoscope. (**A**) Pre-operative CT scan showed SAH. (**B**) Pre-operative CTA scan showed an irregular aneurysm of the PCoA. (**C**) An anterior clinoid process (ACP) was removed under the neuroendoscope before dura mater opening. (**D**) The aneurysm and the branches of the ICA were exposed under the neuroendoscope. (**E**) The aneurysm was clipped under the neuroendoscope. (**F**) The aneurysm was cut to confirm complete clipping. (**G**) We carefully checked the operation area before closing the dura matter. (**H**) Postoperative CT view.

**Figure 5 brainsci-13-01544-f005:**
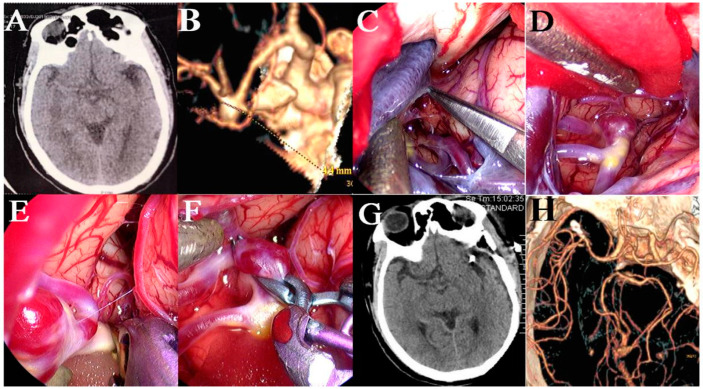
An unruptured MCA aneurysm was clipped via the traditional pterional approach under the neuroendoscope. (**A**) Pre-operative CT scan showed no SAH or hematoma in the brain. (**B**) Pre-operative CTA scan showed a saccular aneurysm at the left MCA bifurcation. (**C**) A sylvian fissure was sharply dissected under neuroendoscope. (**D**) Aneurysm and the branches of MCA were exposed under the neuroendoscope. (**E**) MCA was clipped temporarily. (**F**) Aneurysm was clipped under the neuroendoscope and temporary clip was removed. (**G**) Postoperative CT scan showing no hematoma and damage in the brain. (**H**) Postoperative CTA scan showed that aneurysm was clipped completely and the MCA and its branches were preserved.

**Figure 6 brainsci-13-01544-f006:**
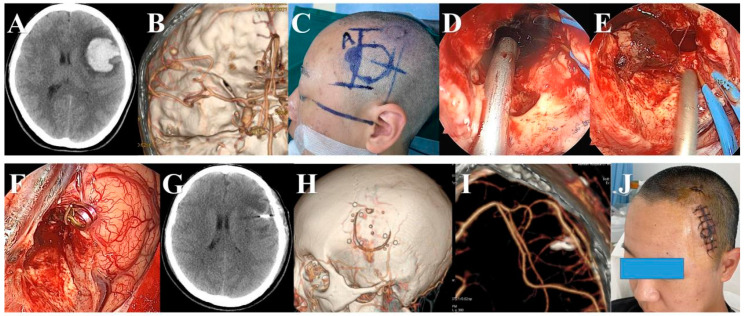
A ruptured M4 segment aneurysm with hematoma was clipped via modified straight skin incision line under the neuroendoscope. (**A**) Pre-operative CT scan showed a hematoma in left frontal lobe. (**B**) Pre-operative CTA scan showed an irregular aneurysm at the left M4 segment. (**C**) A modified straight skin incision line was used. (**D**) Hematoma was evacuated under the neuroendoscope. (**E**) Exposure of aneurysms, parent arteries, and their branches. (**F**) The aneurysm was clipped under the neuroendoscope. (**G**) Postoperative CT view. (**H**) The size of bone flap was displayed via three-dimensional reconstruction of skull CT after operation. (**I**) The aneurysm clip located at the M4 segment and the parental artery was preserved perfectly on postoperative CTA. (**J**) Patient recovered well after operation.

**Figure 7 brainsci-13-01544-f007:**
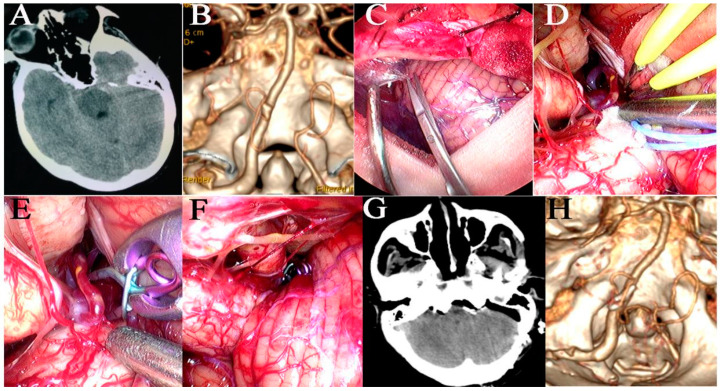
A posterior inferior cerebellar artery (PICA) aneurysm was clipped via the traditional far-lateral surgical approach under the neuroendoscope. (**A**) Mild SAH was mainly located in the posterior fossa on pre-operative CT. (**B**) CTA before the operation showed the aneurysm arising at the origin of the left PICA. (**C**) After opening the dura, the arachnoid membrane was dissected sharply under the neuroendoscope. (**D**) The left VA, inferior loop of the PICA, accessory nerve, and medulla were identified after lifting the cerebellum. (**E**) A mini titanium aneurysm clip was used to clip the aneurysm neck after carefully examining the neck of aneurysm. (**F**) When the aneurysm was clipped, then the structures in the lateral perimedullary cistern were carefully checked. (**G**) The aneurysm clip was shown to have proper placement after the operation. (**H**) The parental artery was preserved perfectly on postoperative CTA.

**Figure 8 brainsci-13-01544-f008:**
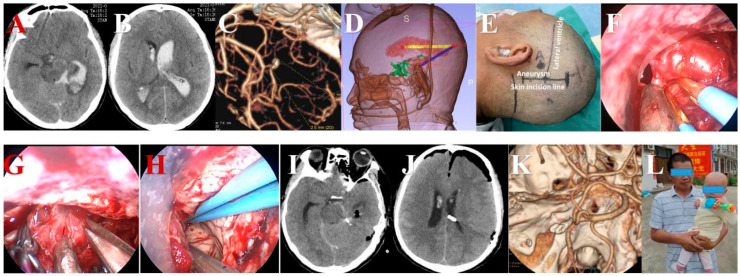
A P2 segment aneurysm was clipped via a modified small skin incision line under the neuroendoscope. (**A**) Preoperative CT showed subarachnoid hemorrhage in the circular cistern. (**B**) Preoperative CT showed hematoma in lateral ventricles. (**C**) Preoperative CTA showed that the aneurysm was located at the P2 segment. (**D**) Lateral ventricles were displayed via three-dimensional reconstruction of skull CT before operation. (**E**) Modified small skin incision line was used. (**F**) Aneurysm, parent arteries, and their branches were exposed. (**G**) Aneurysm was clipped and cut under the neuroendoscope. (**H**) Hematoma in lateral ventricles was evacuated. (**I**) The aneurysm clip was shown to have proper placement after the operation. (**J**) Postoperative CT showed that hematoma in lateral ventricles was removed. (**K**) The aneurysm clip located at the P2 segment and the parental artery was preserved perfectly on postoperative CTA. (**L**) Patient recovered well after operation.

**Table 1 brainsci-13-01544-t001:** Characteristics of patients in the neuroendoscopic and microscopic groups.

Characteristics	Neuroendoscopic Group (*n* = 123)	Microscopic Group (*n* = 106)	Total (*n* = 229)	*p* Value
Gender, *n* (%)				0.045
Male	54 (43.9%)	60 (56.6%)	114 (49.8%)	
Female	69 (56.1%)	46 (43.4%)	115 (50.2%)	
Age, y, mean ± sd	58.504 ± 7.4837	59.5 ± 7.4645	58.965 ± 7.47	0.316
Locations of aneurysms, *n* (%)				0.794
MCA (M1-M4)	40 (32.5%)	29 (27.4%)	69 (30.1%)	
PCoA	38 (30.9%)	34 (32.1%)	72 (31.4%)	
ACoA	27 (22%)	22 (20.8%)	49 (21.4%)	
A2-A4	10 (8.1%)	9 (8.5%)	19 (8.3%)	
PICA	5 (4.1%)	6 (5.7%)	11 (4.8%)	
Opht	3 (2.4%)	6 (5.7%)	9 (3.9%)	
Aneurysm diameter, mean ± sd	9.15 ± 1.9521	9.5167 ± 1.6403	9.31 ± 1.81	0.129
Hunt–Hess grade, *n* (%)				0.435
0	13 (10.6%)	7 (6.6%)	20 (8.7%)	
I	23 (18.7%)	25 (23.6%)	48 (21%)	
II	54 (43.9%)	49 (46.2%)	103 (45%)	
III	18 (14.6%)	18 (17%)	36 (15.7%)	
IV	15 (12.2%)	7 (6.6%)	22 (9.6%)	
Number of bleeding instances, *n* (%)				0.254
0	13 (10.6%)	8 (7.5%)	21 (9.2%)	
1	87 (70.7%)	85 (80.2%)	172 (75.1%)	
≥2	23 (18.7%)	13 (12.3%)	36 (15.7%)	

Abbreviations: MCA= middle cerebral artery, PCoA = posterior communicating artery, ACoA = Anterior communicating artery, Opht = ophthalmic artery, PICA = posterior inferior cerebellar artery.

**Table 2 brainsci-13-01544-t002:** Complications and prognoses of patients in the two groups.

Characteristics	Neuroendoscopic Group (*n* = 123)	Microscopic Group (*n* = 106)	Total (*n* = 229)	*p* Value
Inadequately clipped aneurysms				0.095
No	123 (100%)	102 (96.2%)	225 (98.2%)	
Yes	0 (0%)	4 (3.8%)	4 (1.8%)	
Parent vessel occlusion				0.669
No	123 (100%)	104 (98.1%)	227 (99.1%)	
Yes	0 (0%)	2 (1.9%)	2 (0.9%)	
Surgical procedure time (minutes)	181 (167.5, 192.5)	154 (138, 169)	168 ± 25.61	<0.001
mRS score, *n* (%)				0.277
0~1	90 (73.2%)	83 (78.3%)	173 (75.5%)	
2~3	29 (23.6%)	17 (16%)	46 (20.1%)	
4~5	4 (3.3%)	6 (5.7%)	10 (4.4%)	
Limb hemiplegia, *n* (%)				0.601
No	105 (85.4%)	93 (87.7%)	198 (86.5%)	
Yes	18 (14.6%)	13 (12.3%)	31 (13.5%)	
Hydrocephalus, *n* (%)				0.537
No	110 (89.4%)	92 (86.8%)	202 (88.2%)	
Yes	13 (10.6%)	14 (13.2%)	27 (11.8%)	
Vision loss, *n* (%)				0.082
No	122 (99.2%)	100 (94.3%)	222 (96.9%)	
Yes	1 (0.8%)	6 (5.7%)	7 (3.1%)	
Intracranial infection, *n* (%)				0.300
No	120 (97.6%)	106 (100%)	226 (98.7%)	
Yes	3 (2.4%)	0 (0%)	3 (1.3%)	
Cerebral ischemia, *n* (%)				0.034
No	115 (93.5%)	90 (84.9%)	205 (89.5%)	
Yes	8 (6.5%)	16 (15.1%)	24 (10.5%)	

## Data Availability

The datasets used and/or analyzed during the current study are available from the corresponding author on reasonable request.

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
