# Peer review of "Surgical Clipping of Intracranial Aneurysms Using a Transcranial Neuroendoscopic Approach"

_brainsci, 2023, doi:10.3390/brainsci13111544_

Round 1
Reviewer 1 Report
Comments and Suggestions for Authors
Dear Authors,
I am glad to have the opportunity to review your work. This study aimed to evaluate the feasibility and safety of surgically clipping intracranial aneurysms by transcranial neuroendoscopic approach.
The paper has several shortcomings. In the methodology section, you present the microsurgical clipping as the best method for intracranial aneurysms even though endovascular treatment gives very low complication and mortality rate. The paragraph on that as comparison to the method you are investigating needs to be added. Also, feasibility data on the chosen method from other studies needs to be described. You need to say the name of the institution, to describe who performed the intervention and describe it in general. And to add a paragraph on the statistical analysis method.
The results section needs improvement too. You are investigating safety only by investigating death outcome. I suggest to include modified Rankin score and to also investigate complications rate. All of this needs to be presented in the Results.
English needs to be improved, for example in Abstract: “but 1 patient dead at hospital…”
I suggest major revision.
Comments on the Quality of English LanguageEnglish needs to be improved
Reviewer 2 Report
Comments and Suggestions for Authors
Clarifying some complexities surrounding the surgical clipping of intracranial aneurysms and inproving the surgery outcomes using a transcranial neuroendoscopic approach can be of great importance to the readers of this journal. The authors collected data from more than 128 patients. They claimed that a neuroendoscope can provide clear visualization and a multi-angle view during aneurysm clipping, which is helpful in ensuring adequate clipping and preventing complications. They also reported lower mortality and disability rates and a higher rate of excellence and good outcomes, demonstrating that this technique is safe and reliable. The introduction, results, and methods sections should be completed. The authors did not employ any statistical analysis, and it is unclear how they arrived at these findings.
Comments:
1. What does 'successful clipping' mean in the abstract section? Are you referring to outcomes?
2. The abstract section lacks numerical findings from this paper. I believe it is unnecessary to include details such as 'No patients died during the perioperative period, but one patient died at the hospital due to severe pneumonia and multiple organ failure seven days after the operation. Eleven patients discontinued treatment, and eight patients had poor recovery. The remaining 108 cases recovered well, resulting in an 84.4% good/excellent rate.' Instead, please provide numerical findings.
3. References 1-6 appear outdated for referencing statistics; these statistics may have changed recently. Therefore, it is advisable to use more recent papers as references for 1-6.
4. The last paragraph of the introduction section lacks clarity for readers. You should further clarify your hypothesis and the tools you intend to use to test its validity.
5. The introduction section is rather brief. It is essential to provide a comprehensive overview for readers, explaining the complexities of cerebral aneurysms, variations in surgical outcomes related to aneurysm size (e.g., specific complexities and different outcomes for small-sized aneurysms in surgery ) and the impact of different geometrical complexities on cerebral aneurysm outcomes (e.g., reference 10.1134/S0021894417060025). Additionally, you should highlight safety concerns addressed by the transcranial neuroendoscopic approach (e.g., reference 10.1007/s10143-020-01316-0) and provide an overview of the broader applications of the transcranial neuroendoscopic approach in neuroscience (e.g., reference 10.1038/s41598-021-90927-8). Gradually, lead the readers to the main research question, rather than abruptly transitioning to your final goal without adequate introduction.
6. The inclusion and exclusion criteria are incomplete. This section should be thoroughly detailed.
7. Some subgroups, such as IVH and location P2, include only one patient. Does this make sense? How can you assess the impact of this specific factor on your results? Were the p-values for the sunsections significants?
8. A statistical analysis section should be added to the Methods section.
9. The Results section should present numerical findings, not just describe the dataset. You only descibed the details of your dataset.
10. In the first paragraph of the Discussion section should be relocated to the Introduction section.
11. I find it challenging to understand this paper. How did you derive the outcomes mentioned, such as "...neuroendoscope can provide clear visualization and a multi-angle view during aneurysm clipping, which is helpful in ensuring adequate clipping and preventing complications.' or 'The lower mortality and disability rate and higher excellence and good rate show that this technique is safe and reliable..."? Your results do not seem to support these findings.
12. It is puzzling that you did not apply any analytical methods. How did you arrive at these conclusions? To report results, you need to employ statistical analysis. Without using statistics, it seems more like a report than a research paper.
13. The main concern with this paper is its analysis and methodology. The authors lack sufficient evidence to support their main claims, and it is unclear how they arrived at these results.
14. Mentioning "lower mortality and disability rate and higher excellent and good rate" as the main findings with only 128 patients seems inadequate. To discuss rates effectively, a much larger dataset is required. Typically, for rates, researchers use big data, and this size of dataset may not adequately support rate-based conclusions.
15. You used sensitive words like 'safe' and 'reliable' and so on without heeding the important note that these are critical concepts. When using such terms, it is essential to consider various statistical considerations, as they are not general words to be employed freely in this paper. There is not any statistical evedences but you use these sensitive words!?
16. The level of English language proficiency needs improvement.
Comments on the Quality of English Language
moderate level
Reviewer 3 Report
Comments and Suggestions for Authors
The paper requires major revisions. However, to further evaluate and clarify some aspects of the study, I would like to ask the following 24 questions:
1. Can you provide more details about the methodology used in this retrospective study?
2. What were the specific inclusion and exclusion criteria for selecting the 128 patients?
3. How were the patients with intracranial aneurysms treated differently using full neuroendoscopy (FNE) compared to neuroendoscopy combined microsurgery (ECM)?
4. Were there any specific surgical techniques or modifications that were employed during the procedures?
5. Can you elaborate on the types of complications that the 22 patients experienced and how they were managed?
6. What was the rationale behind employing the neuroendoscope in this study for visualizing regional vascular anatomy?
7. How did the neuroendoscope help in assessing the aneurysmal neck and its relationship with the parent artery?
8. Were there any challenges or limitations encountered when using the neuroendoscope during the surgeries?
9. Can you provide more information about the criteria used to assess the outcomes of the procedures?
10. Were there any specific factors that influenced whether a patient received full neuroendoscopy or neuroendoscopy combined microsurgery?
11. How did the presence of hematoma or subarachnoid hemorrhage influence the surgical approach and outcomes?
12. Were there any complications directly related to the use of the neuroendoscope itself?
13. What percentage of aneurysms were completely clipped, and were there any cases of residual necks or incomplete clipping?
14. Can you discuss the criteria used for choosing between microsurgery and neuroendoscopy during the procedures?
15. Were there any significant differences in outcomes between patients treated with full neuroendoscopy and those treated with neuroendoscopy combined microsurgery?
16. How were the patients monitored postoperatively for complications or recurrence?
17. Can you explain the reasons behind the decision of some patients or their families to give up treatment?
18. Were there any specific factors or patient characteristics associated with a higher risk of complications or poorer outcomes?
19. Did the surgeon's experience or expertise with neuroendoscopy play a significant role in the outcomes?
20. Can you explain the clinical implications of your findings for oncologists and patients, especially in terms of treatment decision-making? How do the findings from this study compare with the previous AI studies mentioned in the literature review section? Add suitable reference to the site with PMID: 37238175].
21. Were there any instances where the neuroendoscope revealed unexpected findings that influenced the surgical approach?
22. Can you provide more information about the patient demographics and the distribution of aneurysm locations?
23. How were the outcomes measured in terms of Glasgow Coma Scale (GCS) and the GE rate?
24. What are the future directions or plans for further research in this area, considering the need for more data support mentioned in the conclusion?
Round 2
Reviewer 1 Report
Comments and Suggestions for Authors
Dear authors, thank you for revising the paper. It has much improved.
Author Response
Thank you for your feedback. I am glad that you are pleased with the revisions we have made.
Reviewer 2 Report
Comments and Suggestions for Authors
Thank you for the correction. This version is much better than the original one. I have still concerns related to the introduction (Comment 5) and discussion. You missed mentioning some important points in the introduction section:
1. In several instances, you mention that this study aims to evaluate the "feasibility and safety" in intracranial aneurysms. Therefore, it is crucial to establish a strong background about these concepts in the introduction section or discuss and compare related papers in the discussion section (introduction or discussion is your option). This paper (10.1007/s10143-020-01316-0) can be useful in this regard.
2. Since one of your main conclusions is "we can have a good view of the aneurysm neck, which helps assess the effectiveness of aneurysm clamping," it is essential to discuss enough background related to similar papers about the effects of the "aneurysm neck" on rupture risk or treatment outcomes. You should clarify the role of the aneurysm neck in rupture risk and consequently in treatment outcomes first (10.1134/S0021894417060025) to highlight the importance of your main finding for the practical use of clinicians. First, you should say previous studies highlighted this problem and you tried to answer it.
Reviewer 3 Report
Comments and Suggestions for Authors
The article revised by the author according to the comments of the reviewers basically meets the requirements.
Author Response

(The authors gave the same response as above.)
